# Comparisons of retinal vessel density and glaucomatous parameters in optical coherence tomography angiography

**Zhen Li◉\*, Zhike Xu, Qiang Liu, Xiaoli Chen, Linrui Li**

Department of Ophthalmology, The People's Hospital of Leshan, Leshan, Sichuan Province, China

\* lizhen81131@163.com

## Abstract

### Purpose

To compare the retinal vessel density and glaucomatous parameters in primary angle closure glaucoma (PACG), to evaluate the diagnostic and monitoring abilities of the peripapillary and macular vessel density in the progression of glaucoma.

### Methods

This was a observational, prospective and cross-sectional study. According to Glaucoma Staging System, 218 eyes (116 participants) were divided into 5 groups: no glaucoma, early glaucoma, moderate glaucoma, advance glaucoma, severe glaucoma. All participants underwent a comprehensive ocular examination, which included corrected distance visual acuity measurement, slit-lamp biomicroscopy, intra ocular pressure (IOP), gonioscopy, fundus examination, stereoscopic optic disc photography, Humphrey visual field test(VF), peripapillary and macular optical coherence tomography angiography(OCTA) scan. SPSS software was used to calculate and compare retinal vessel density (peripapillary vessel density, PVD and macular vessel density, MVD) and glaucomatous parameters (mean deviation (MD),pattern standard deviation(PSD), retinal nerve fiber layer (RNFL), ganglion cell-inner plexiform layer(GCIPL),rim area, average cup/disc(C/D) ratio).

### Results

The GCIPL thickness, RNFL thickness, PVD and MVD are significantly reduced in PACG. There were significant differences in all measurements among the groups (P<0.01). Reduced peripapillary and macular vessel density in glaucoma were detected and a statistically significant correlation with glaucoma stages (P<0.01). In addition, the results of retinal vessel density, reduced RNFL thickness and GCIPL thickness were also statistically related to the stage of glaucoma. As expected, the rim area was significantly smaller with higher C/D area ratios in glaucomatous eyes corresponding to the severity of disease.

**Data Availability Statement:** All relevant data are within the manuscript and its Supporting Information files.

**Funding:** This study was supported by the science and technology foundation of Sichuan provincial health and family planning commission (NO: 190065). The funders had no role in study design, data collection and analysis, decision to publish, or preparation of the manuscript.

**Competing interests:** The authors have declared that no competing interests exist.

**Abbreviations:** PACG, primary angle closure glaucoma; POAG, primary open angle glaucoma; IOP, intra ocular pressure; VF, visual field; OCTA, optical coherence tomography angiography; PSD, peripapillary vessel density; MVD, macular vessel density; MD, mean deviation; PSD, pattern standard deviation; RNFL, retinal nerve fiber layer; GCIPL, ganglion cell-inner plexiform layer; GC, ganglion cell; SAP, standard achromatic perimetry; FAZ, foveal avascular zone.

## Conclusions

The changes of PVD and MVD had strongly positive correlation with GCIPL thickness and RNFL thickness, had negative correlation with the severity of glaucoma, which meant the more severe the glaucoma was, the lower PVD and MVD were. Compared to traditional glaucoma staging system judged by VF, the changes of PVD and MVD obtained by OCTA might be a new method to grade the stage of glaucoma. These findings theorize that the changes of PVD and MVD may be better facilitated for the observation and monitoring of glaucoma progression.

## Introduction

Glaucoma is the leading cause of irreversible blindness in the world, around 65 million people suffer from this disease. [1] It is well documented that early diagnosis, early treatment and monitoring of the progression are the key determinants to reduce the risk of irreversible vision loss.

Though, the pathogenesis of glaucoma is not fully understood, reducing IOP is still the only effective method to slow glaucoma damage [2].As we know, structural damage occurs prior to functional damage in glaucoma. Functional, such as VF may be not appropriate to diagnose the earliest stages of glaucoma or judge the progression of glaucoma. Because of a significant amount of ganglion cell (GC) loss is expected to occur (25–50%) before functional deficits[3]. Peripapillary retinal nerve fiber layer (PRNFL) thinning, GC thickness reduction and neuro-retinal rim narrowing are the available good standards for diagnosing and monitoring the progression of glaucoma, while detection of VF defects is indispensable for monitoring the functional decline.

With the recent development of OCTA, which can provide us with qualitative and quantitative information of the microvasculature in various retinal regions, including the optic nerve, peripapillary retina, and macula, we can easily get insight of the relationship between retinal blood flow various and visual functional damage. Some studies have recently suggested that reduced ocular blood flow is a primarily independent metric of visual function outside of other structural parameters, supporting a vascular role in the development of glaucoma [4–6]. In the past few years, some research has shown decreased peripapillary retinal and macular perfusion in glaucomatous eyes correlating with VF damage [7–8].These studies however focused more on primary open angle glaucoma (POAG). Due to anatomical variations, the prevalence of primary angle closure glaucoma (PACG) is greater than POAG in Asia. The main purpose of this study was to compare the retinal vascular density and glaucomatous parameters in PACG and to evaluate the diagnostic and administrative abilities of the peripapillary and macular vessel density in the progression of glaucoma.

## Methods

This was a observational, prospective, cross-sectional study at the department of Ophthalmology in The People's Hospital of Leshan between January 2019 and February 2020. The research protocol was approved by the Ethics Committee of People's Hospital of Leshan and adhered to the Declaration of Helsinki. Written informed consents were obtained from all participants. Participants of this study included normal control subjects (11 person, 20 eyes)and chronic PACG patients(105 patients, 198 eyes). According to Glaucoma Staging System, all subjects

were divided into 5 groups: no glaucoma, early glaucoma, moderate glaucoma, advance glaucoma, severe glaucoma. Normal subjects had no family history of glaucoma, intraocular pressure (IOP)≤21mm Hg, open angles on gonioscopy, normal anterior and posterior segment on slit-lamp examination, no glaucomatous optic discs, no RNFL defects, and no glaucomatous VF defects. A normal VF was defined as a PSD within the 95% confidence limits and a Glaucoma VF test within normal limits. PACG patients had closed angles on gonioscopy. PACG patients had glaucomatous changes on ONH examination and VF examination, including rim narrowing, notching, RNFL defects. Glaucomatous VF defects were defined as a cluster of ≥ 3 points with P < 0.05 on the pattern deviation map in at least one hemifield, including ≥ 1 point with P < 0.01; a PSD of P < 0.05; or glaucoma VF test result outside the normal limits. Exclusion criteria were presence of any retinal or macular pathology, any media opacities, and any systemic or neurological conditions that could produce VF defects. Patients unable to perform reliable VF testing or with poor quality OCTA, including segmentation errors, were also excluded. All participants underwent a comprehensive ocular examination, which included corrected distance visual acuity measurement, slit-lampbiomicroscopy,Goldmann applanation tonometry, gonioscopy, fundus examination, stereoscopic optic disc photography.

## Humphrey visual field test

VF test was performed with the Swedish Interactive Threshold Algorithm (SITA) Standard strategy program 24–2 of the Humphrey Field Analyzer 3 (Carl Zeiss Meditec, Jena, Germany). Fixation losses ≤20%, false positives, and false negatives ≤33% were established as the reliability criteria. The glaucoma stage was determined based on Glaucoma Staging System (GSS)[9]. In brief, the GSS stage assignment was based primarily on Humphrey visual field parameters. The value of MD determined the stage of severity: stage 0 (no glaucoma) was determined when MD is of more than -1.0 dB, stage 1 (early glaucoma) with MD between -1.00 dB and -5.00 dB, stage 2 (moderate glaucoma) with MD between -5.01 dB and -12.00 dB, and stage 3 (advance glaucoma) with MD is between -12.01 dB and -20Db,stage 4 (severe glaucoma) with MD is more than -20 dB.

## Structural parameters and vessel density

The images of Macular and peripapillary 6x6 mm scans were acquired with SD-OCT (Cirrus, HD-OCT 5000; Carl Zeiss Meditec, Inc.) to get GCIPL thickness(calculates the GCIPL thickness measurements in 6 sectors of the macula), peripapillary RNFL thickness, Rim area, C/D ratio, MVD and PVD. OCTA scans with a signal strength > 7.Angioplex Metrix software of the Cirrus HD-OCT automatically calculates 2 parameters from the superficial retinal layer slab: vessel length density, defined as the total length of perfused vasculature per unit area in the region of measurement, and perfusion density, defined as the total area of perfused vasculature per unit area in the region of measurement. In our study, we considered vessel length density for analysis. AngioPlex™ subdivides the scan into four areas: a central circle, an inner circle, an outer circle and a full circle.

## Statistical analysis

All the statistical analyzes were performed using the SPSS software version 20 (SPSS Inc, Chicago). A chi-square test was used to compare different demographic characteristics, including age, gender, history of anti-glaucomatous surgery, and use of anti-glaucomatous medications. Descriptive statistics were used to calculate mean and median of MD, PSD, RNFL,GCIPL, rim area, average C/D, PVD and MVD. Student's t tests were used to compare the average values of measurements between normal and glaucomatous. Univariate analysis with Pearson

correlation test and multivariate analysis with ANOVA testing were performed to determine the correlation between structural parameters and functional parameters, vessel density and functional parameters. Significance was taken as p < 0.05. However, for multiple comparisons among groups, a Kruskal-Wallis test with Dunn's correction was applied with resultant significance level set at p < 0.01.

## Results

### Demographic characteristics of glaucoma and normal subjects

The mean of age was 61.3±19.0 for men and 63.6±14.8 for women.Compared to the normal group, MD was significantly lower with higher IOP and PSD in glaucoma group.(Table 1)

### The changes of functional parameters, structural parameters and retinal vessel density

Table 2 showed the changes of functional parameters, structural parameters and retinal vessel density among the five groups (normal, early, moderate, advance and severe group). There were significant differences in all measurements among the groups, including the functional parameters (MD and PSD), structural parameters(average, superior and inferior areas of RNFL thickness, GCIPL thickness), rim area, average C/D ratio, PVD and MVD (central, inner, outer, full areas)(p < 0.01).From Table 2, we could find that the average PVD(full area) was 18.50±0.52 $mm^{-2}$(from 18.00 to 19.30 $mm^{-2}$), while the average MVD (full area) was 17.75 ±0.44 $mm^{-2}$(from 17.20 to 18.40 $mm^{-2}$) in the normal group. The average PVD (full area) was 15.83±2.11 $mm^{-2}$(from 12.70 to 18.00 $mm^{-2}$), while the average MVD (full area) was 15.19 ±2.26 $mm^{-2}$(from 10.60 to 17.60 $mm^{-2}$) in the early group. The average PVD (full area) was 14.46±2.29 $mm^{-2}$(from 10.20 to 17.80 $mm^{-2}$), while the average MVD(full area) was 13.50 ±2.93 $mm^{-2}$(from 8.20 to 17.50 $mm^{-2}$) in the moderate group. The average PVD (full area) was 12.19±3.87 $mm^{-2}$(from 6.00 to 17.10 $mm^{-2}$), while the average MVD (full area) was 11.78±4.87 $mm^{-2}$(from 3.10 to 17.30 $mm^{-2}$) in the advance group. The average PVD (full area) was 10.48 ±3.64 $mm^{-2}$(from 4.80 to 15.10 $mm^{-2}$), while the average MVD (full area) was 10.89±3.46$mm^{-2}$(from 3.00 to 16.10 $mm^{-2}$) in the severe group.

**Table 1. Demographic characteristics of the study subjects.**

|  | Normal (20 eyes, 11 patients) | Glaucoma (198 eyes, 105 patients) | PP Value |
|---|---|---|---|
| Age (y) | 68.2 ± 1.2 | 70.1 ± 0.8 | 0.25 [a] |
| Gender, n (Male/Female) | 8/3 | 68/37 | 0.08 [b] |
| VF |  |  |  |
| MD (dB) | -0.247 ± 0.135 | -12.586 ±10.192 | < 0.0001 [a] |
| PSD (dB) | 1.460 ± 0.167 | 6.706 ± 4.039 | < 0.0001 [a] |
| IOP (mmHg) | 16.500 ±1.539 | 25.354 ±9.380 | 0.001 [a] |
| Topical glaucoma medication, N |  |  |  |
| 0 | null | 55 |  |
| 1 | null | 48 |  |
| >1 null | null | 36 |  |
| Glaucoma surgery history, N | null | 72 |  |

dB = decibels; IOP = intraocular pressure; MD = mean deviation; PSD = pattern standard deviation VF = visual field.

[a] Statistical significance tested with Student's t test.

[b] Statistical significance tested with the Chi-square test.

**Table 2. Results of functional parameters, structural parameters and vessel density.**

| Variables | Normal | Early | Moderate | Advance | Severe | P [a] |
|---|---|---|---|---|---|---|
| (group1,n = 20 eyes) | (group2,n = 70 eyes) | (group3,n = 46 eyes) | (group4,n = 31 eyes) | (group4,n = 51) | Value | |
| **Functional parameters (VF)** | | | | | | |
| MD(dB) | -0.25±0.14 | -2.49±1.06 | -9.35±1.73 | -15.31±1.82 | -27.70±3.62 | P<0.01 |
| PSD(dB) | 1.46±17 | 2.32±1.04 | 7.56±3.26 | 8.73±1.74 | 10.73±2.17 | P<0.01 |
| **Structural parameters** | | | | | | |
| Average RNFL thickness (μm) | 102.75±9.77 | 79.26±12.23 | 67.89±9.06 | 65.87±10.51 | 61.94±9.54 | P<0.01 |
| Superior RNFL thickness (μm) | 129.00±14.23 | 98.50±19.28 | 84.70±20.07 | 72.35±12.93 | 69.61±9.8 0 | P<0.01 |
| Inferior RNFL thickness (μm) | 142.25±16.02 | 95.26±20.35 | 74.70±14.34 | 72.13±20.54 | 64.37±10.09 | P<0.01 |
| Average GCL+IPL thickness (μm) | 87.75±1.83 | 67.43±7.93 | 66.91±8.69 | 59.52±6.57 | 51.88±3.64 | P<0.01 |
| Superior GCL+IPL thickness (μm) | 89.00±2.18 | 68.07±9.88 | 66.39±14.93 | 62.35±12.34 | 55.84±6.19 | P<0.01 |
| Inferior GCC+IPL thickness (μm) | 85.25±3.93 | 65.83±8.35 | 63.53±11.61 | 58.52±6.59 | 56.37±5.11 | P<0.01 |
| **Optic disc** | | | | | | |
| Rim area (mm 2) | 1.30±0.25 | 1.17±0.44 | 0.81±0.21 | 0.75±0.33 | 0.43±0.12 | P<0.01 |
| Average C/D ratio | 0.50±0.19 | 0.61±0.22 | 0.78±0.31 | 0.82±0.07 | 0.90±0.05 | P<0.01 |
| **Macular vessel density (1/mm$^2$)** | | | | | | |
| Central | 8.48±0.83 | 7.11±2.81 | 6.05±2.72 | 5.79±3.55 | 5.04±3.41 | P<0.01 |
| Inner | 17.80±0.58 | 15.53±2.33 | 14.67±3.21 | 12.48±6.72 | 12.04±4.31 | P<0.01 |
| Outer | 18.05±0.43 | 15.38±2.34 | 13.40±2.94 | 12.20±4.46 | 10.94±3.48 | P<0.01 |
| Full | 17.75±0.44 | 15.19±2.26 | 13.50±2.93 | 11.78±4.87 | 10.89±3.46 | P<0.01 |
| **peripapillary vessel density(1/mm$^2$)** | | | | | | |
| Central | 1.01±1.62 | 4.60±4.06 | 3.97±5.65 | 1.86±3.59 | 0.12±0.17 | P<0.01 |
| Inner | 18.15±0.92 | 15.47±2.10 | 13.49±2.26 | 8.73±5.05 | 8.20±3.33 | P<0.01 |
| Outer | 19.18±0.39 | 16.40±2.96 | 15.44±2.58 | 13.58±3.67 | 11.44±4.05 | P<0.01 |
| Full | 18.50±0.52 | 15.83±2.11 | 14.46±2.29 | 12.19±3.87 | 10.48±3.64 | P<0.01 |

GCL+IPL = ganglion cell layer+inner plexiform layer; MD = mean deviation; PSD = pattern standard deviation; C/D = cup/disc; RNFL = retinal nerve fiber layer;

VF = visual field.

Numbers displayed are mean ± standard deviation.

[a] Differences between groups were tested with Kruskal-Wallis test with Dunnett's correction.

In the en face OCTA image of the superficial vascular complex (SVC), there are centripetally branching vessels terminating in the central foveal avascular zone (FAZ). The branching vessels were observably denser in normal eyes than those with glaucoma (Fig 1A). The changes of PVD were the same as MVD (Fig 1B). Reduced PVD and MVD in glaucoma were detected and a statistically significant correlation with glaucoma stages (Fig 1C) was appreciated. The data confirmed a correlation between retinal vessel density and glaucomatous stage. Fig 2 showed the changes of RNFL and GCIPL thickness among five groups (normal, early, moderate, advance and severe groups).The GCC is absent at the fovea, but gradually becomes thicker and reaches the thickest point at the parafoveal annulus. As expected, the rim area had significantly with higher C/D area ratios in glaucomatous eyes corresponding to the severity of disease (Fig 2A). Reduced RNFL thickness and GCIPL thickness were detected and with a statistical relation to the stage of glaucoma, which is in agreement with the traditional diagnosis of glaucoma.(Fig 2B).

## Correlation of retinal vessel density with structural and functional tests

Table 3 highlighted the correlation of retinal vessel density with structural and functional parameters in glaucomatous eyes.PVD (central, inner, outer, full) and MVD (central, inner,

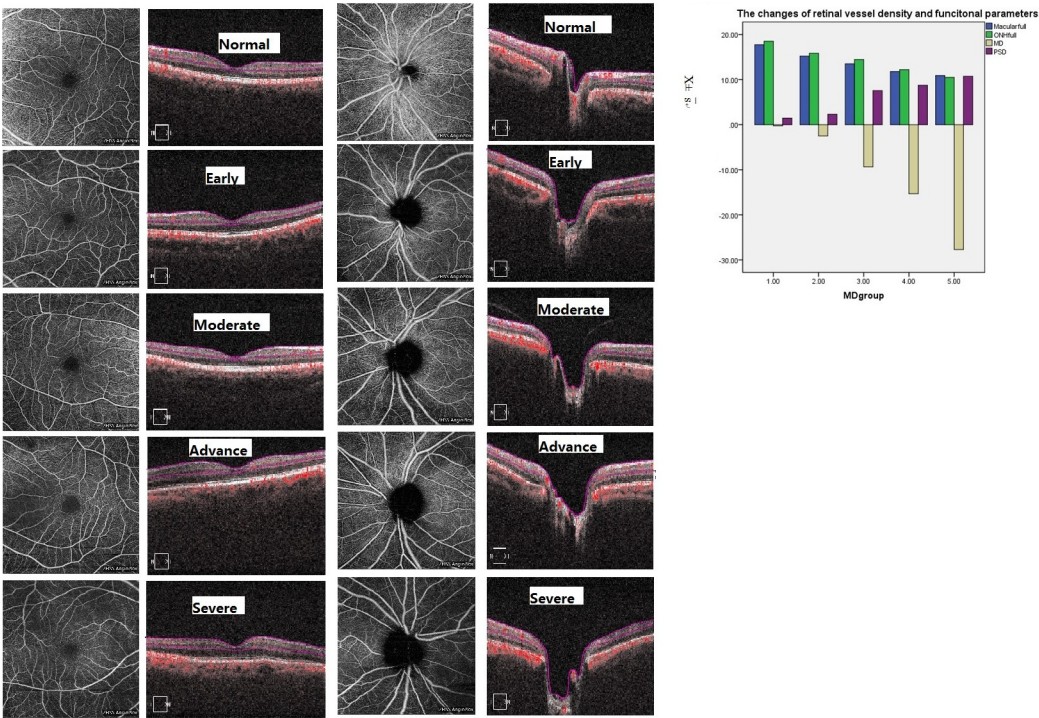

**Fig 1. The changes of peripapillary and vascular perfusion in different stages of glaucoma.** A. These are macular vascular images of 6 x6 mm en face angiograms generated by OCTA. Images in the first column are AngioPlex-superficial maps. The second column are the corresponding OCT sections of retina. From the images, we can see that a significant reduction of MVD in glaucoma eyes compared to normal eyes. B. These are peripapillary vascular images of 6 x6 mm en face angiograms generated by OCTA. Images in the first column are AngioPlex-ONH maps. The second column are the corresponding OCT sections of ONH From the images, we can see that a significant reduction of PVD in glaucoma eyes compared to normal eyes. C. The bar graphs demonstrate that the reduced MVD and PVD are strongly related to the severity of glaucoma, the more severe the glaucoma was, the lower PVD and MVD were.

outer, full) showed a strong positive correlation with MD and rim area, while negatively correlated with PSD and average C/D ratio (P≤0.01). Perpapillary vessel density(central, inner, outer, full) were positively correlated with GCIPL and RNFL thickness(average, superior, inferior)(p≤0.01). MVD (central, inner, outer, full) were positively correlated with GCIPL thickness (average, superior, inferior).There was no correlation between the MVD and RNFL thickness in the central(r = -0.039 p = 0.583) and inner (r = 0.08 p = 0.916)areas of the macula.

## Discussion

Prior studies have shown that glaucomatous damage can be detected by OCT as reduced RNFL thickness in the peripapillary region or thinning of the ganglion cell complex within the macula area[10–11].With the development of OCTA, which now can provides us with qualitative and quantitative information of various retinal regions.

In our study, we also proved that the reduced PVD and MVD are significantly correspond with the severity of VF damage along the glaucoma severity stages, regardless of the state of anterior chamber angle, which was agreement with some prior researches [12–15].We also found that reduced RNFL thickness and GCIPL thickness were detected and with a statistical relation to the stage of glaucoma, which is agree with the traditional diagnosis standard of glaucoma. In the initial phase of glaucoma, functional damage might be undetectable with VF testing, but the structural damage had been detected, with a loss of retinal ganglion cells. Some

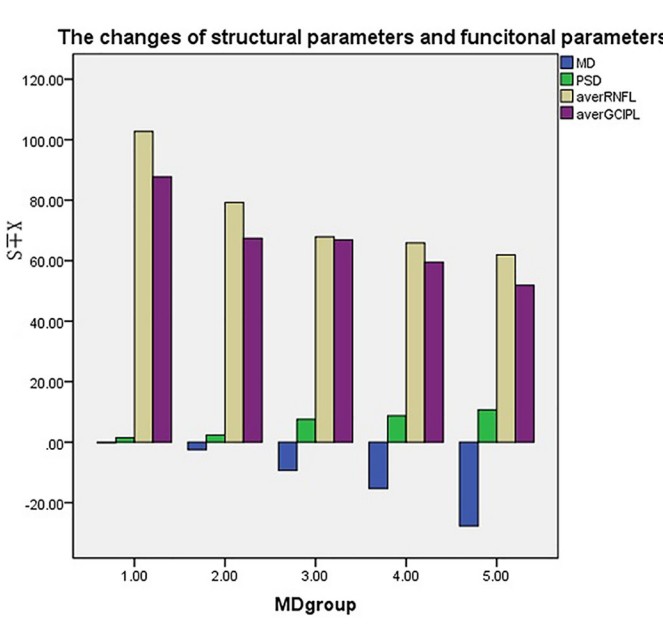

**Fig 2. The changes of RNFL, GCIPL thickness and VF in different stages of glaucoma.** A. Combined GCA and RNFL deviation maps are in the first column, the maps of RNFL thickness are in the second column, GCIPL are in the third column, the corresponding VF maps are in the fourth column. From the maps, we can see that a significant reduction of RNFL and GCIPL thickness. B. The bar graphs demonstrate that the reduced RNFL and GCIPL thickness are strongly related to the severity of glaucoma, the more severe the glaucoma was, the thinner RNFL and GCIPL thickness were.

studies had shown the structure-function relationships between various glaucoma parameters and severity in visual field testing [16–17].

Additionally, we had shown the positive correlation of PVD and MVD with MD and rim area, while negatively correlated with PSD and average C/D ratio, which means the more severe the glaucoma was, the lower PVD and MVD were. Wu et al [18] consider reduced MVD occurs in POAG despite of age-related changes, which also correlates with reductions in RNFL and GCC measurements. In our study, Table 3 showed that PVD(central, inner, outer, full) were positively correlated with GCIPL and RNFL thickness(average, superior, inferior). MVD (central, inner, outer, full) were positively correlated with GCIPL (average, superior, inferior), but with no correlation to the RNFL thickness in the central and inner area of macula. Manalastas et al [19] consider that the weaker association of MVD compared with ONH vessel density with tissue thickness may be due to differences in micorovasculature between the macula and ONH. It is well documented that GC is absent at the fovea of macular, then gradually becomes thicker and reaches the thickest point at the parafoveal annulus, which may explain why MVD in the central and inner area of macula has no correlation with RNFL.

Rao et al [20] suggested that structural changes in PACG occurred earlier than the reduction in retinal vessel density. But, Richter et al [21] considered that OCTA parameters had stronger associations with functional rather than structural measures of glaucoma. It has long been debated whether reduced blood perfusion in glaucomatous eyes is secondary to loss of

**Table 3. The correlation of retinal vessel density with structural and functional tests in glaucomatous eyes.**

| | Central | inner | outer | full | central | inner | outer | full |
|---|---|---|---|---|---|---|---|---|
| **Glaucoma (n = 198)** | | | | | | | | |
| VF | | | | | | | | |
| MD(dB) | r = 0.303** | r = 0.685** | r = 0.506** | r = 0.586** | r = 0231*** | r = 0.281** | r = 0.436** | r = 0.442 ** |
| PSD(dB) | r = -2.53** | r = -0.525** | r = -0.367** | r = -4.40** | r = -1.23** | r = -1.62* | r = -0.288** | r = -0.267** |
| Disc | | | | | | | | |
| Rim area | r = 0.647** | r = 0.569** | r = 0.192** | r = 0.322** | r = 0.321** | r = 0.184** | r = 0.469** | r = 0.407** |
| Average C/D ratio | r = -0.559** | r = -0.524** | r = -0.288** | r = -0.228*** | r = -0.244** | r = -0.193** | r = -0.408** | r = -0.365** |
| Structural parameters | | | | | | | | |
| Average GCIPL thickness(um) | r = 0.174** | r = 0.664** | r = 0.599** | r = 0.655** | r = 0.294** | r = 0.429** | r = 0.537** | r = 0.533** |
| Superior GCIPL thickness(um) | r = 0.177** | r = 0.513** | r = 0.426** | r = 0.463** | r = 0.295** | r = 0.337** | r = 0.419**1 | r = 0.418** |
| Inferior GCIPL thickness(um) | r = 0.182** | r = 0.419** | r = 0.422** | r = 0.458** | r = 0.232*** | r = 0.303** | r = 0.428** | r = 0.407** |
| Average RNFL thickness (μm) | r = 0.524** | r = 0.412** | r = 0.287** | r = 0.197** | r = -0.039 | r = 0.080 | r = 0.271** | r = 0.225*** |
| Superior RNFL thickness(um) | r = 0.361** | r = 0.541** | r = 0.282** | r = 0.382** | r = -0.030 | r = 0.175* | r = 0.313** | r = 0.294** |
| Inferior RNFL thickness(um) | r = 0.514** | r = 0.372** | r = 0.284 ** | r = 0.186** | r = -0.027 | r = -0.038 | r = 0.266** | r = 0.204** |

RNFL = retinal nerve fiber layer; GCIPLC = ganglion cell-inner plexiform layer complex. MD = mean deviation; PSD = pattern standard deviation; C/D = cup/disc;

RNFL = retinal nerve fiber layer; VF = visual field.

Correlation between parameters was tested with Pearson correlation test.

*P<0.05

**P≤0.01

***P≤0.001.

ganglion cells or reduce of RNFL thickness, but we consider that the glaucoma affects the RGC density in the macula, and the decreased blood supply at the affected area could be the result or consequence of glaucomatous damage. This hypothesis is still worthy of long-term research.

Compared to traditional glaucoma staging system judged by VF, the results of PVD and MVD might be a new glaucoma staging system judged by OCTA. In our study, we attempted to categorize the stage of glaucoma according the average value of MVD and PVD: stage 0 (no glaucoma) was determined when PVD was of more than 18.50 mm$^{-2}$ or MVD was of more than 17.75 mm$^{-2}$; stage 1 (early glaucoma) with PVD between 15.83 and 18.50 mm$^{-2}$ or MVD between 15.19 and 17.75 mm$^{-2}$; stage 2 (moderate glaucoma) with PVD between 14.46 and 15.83 mm$^{-2}$ or MVD between 13.50 and 15.19 mm$^{-2}$; stage 3 (advance glaucoma) with PVD between 12.19 and 14.46 mm$^{-2}$ or MVD between 11.78 and 13.50 mm$^{-2}$; stage 4 (severe glaucoma) with PVD is less than 12.19 mm$^{-2}$ or MVD is less than 11.78 mm$^{-2}$. Of course, this new staging system was based on the average value of MVD and PVD in full area, we still need much more data to support the staging standard or more rational approach to categorize the stage of glaucoma.

There were still some limitations to our study. Firstly, we were planned to recruit more glaucoma subjects but ignored that a smaller number of normal participants were recruited than glaucoma as whole, and therefore there is a potential bias in interpretation. Besides, longitudinal and reproducible study will be required to provide more information of how retinal vessel density and structural changes in glaucoma progression with OCTA.

In summary, the changes of PVD and MVD had strongly positive correlation with GCIPL thickness and RNFL thickness, while had negative correlation with the severity of glaucoma, which meant the more severe the glaucoma was, the lower PVD and MVD were. According to the results of PVD and MVD, we may attempt to grade the staging of glaucoma by OCTA. We suggested that the changes of PVD and MVD might be better facilitated in detecting and monitoring the progression of glaucoma.

## Supporting information

**S1 Data.**
(SAV)

## Acknowledgments

The authors thank Dr Yu Han for technical assistance.

## Author Contributions

**Conceptualization:** Zhen Li.

**Data curation:** Zhike Xu, Qiang Liu, Xiaoli Chen, Linrui Li.

**Investigation:** Qiang Liu.

**Methodology:** Zhen Li, Zhike Xu.

**Writing – original draft:** Zhen Li.

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
