## [Decision Letter · Decision Letter 0]

16 Mar 2020

PONE-D-20-04584

Comparisons of retinal vessel density and glaucomatous parameters in optical coherence tomography angiography

PLOS ONE

Dear Dr. Li,

Thank you for submitting your manuscript to PLOS ONE. After careful consideration, we feel that it has merit but does not fully meet PLOS ONE’s publication criteria as it currently stands. Therefore, we invite you to submit a revised version of the manuscript that addresses the points raised during the review process.

The learned reviewers have offered a number of criticisms that need to be addressed by making appropriate changes in the manuscript. 

We would appreciate receiving your revised manuscript by Apr 30 2020 11:59PM. To enhance the reproducibility of your results, we recommend that if applicable you deposit your laboratory protocols in protocols.io, where a protocol can be assigned its own identifier (DOI) such that it can be cited independently in the future. For instructions see: http://journals.plos.org/plosone/s/submission-guidelines#loc-laboratory-protocols

We look forward to receiving your revised manuscript.

Kind regards,

Sanjoy Bhattacharya

Academic Editor

PLOS ONE

Journal Requirements:

https://www.tandfonline.com/doi/abs/10.1080/02713683.2018.1563195?journalCode=icey20

https://tvst.arvojournals.org/article.aspx?articleid=2718262

https://www.sciencedirect.com/science/article/pii/S0181551218302602?via%3Dihub

In your revision ensure you cite all your sources (including your own works), and quote or rephrase any duplicated text outside the methods section. Further consideration is dependent on these concerns being addressed.

"This study was supported by the science and technology foundation of Sichuan

provincial health and family planning commission (NO:190065). This study was also partially supported by the Innovation Project of Leshan people’s hospital. The funding body had no role in the design or conduct of this study. The funding organizations had no role in the study design, conduct of this research, data analysis, decision to publish, or preparation of the manuscript."

"The authors received no specific funding for this work."

Reviewers' comments:

Reviewer's Responses to Questions

**Comments to the Author**

1. Is the manuscript technically sound, and do the data support the conclusions?

Reviewer #1: Yes

Reviewer #2: Yes

2. Has the statistical analysis been performed appropriately and rigorously? 

Reviewer #1: Yes

Reviewer #2: Yes

3. Have the authors made all data underlying the findings in their manuscript fully available?

Reviewer #1: Yes

Reviewer #2: Yes

4. Is the manuscript presented in an intelligible fashion and written in standard English?

Reviewer #1: No

Reviewer #2: No

5. Review Comments to the Author

Reviewer #1: This study demonstrates the relationship between retinal vessel density (peripapillary vessel density, PVD and macular vessel density, MVD) and glaucomatous parameters (mean deviation(MD),pattern standard deviation(PSD), retinal nerve fiber layer (RNFL), ganglion cell inner plexiform layer (GCIPL),rim area, average cup/disc(C/D) ratio).

Abstract:

Purpose:

1. Line 29: what do you mean by administrative abilities?

Results:

Line 45: “Reduced peripapillary and macular vessel density in glaucoma was

1. 46 detected and a statistically significant correlation with glaucoma stages (P<0.01).”

correction: were

Conclusion:

1. Line 51-55: need English grammar correction, I will reword the conclusion

Introduction:

1. Line 63: “As all we known” you mean as all we know?

2. Line 67: as all we know

3. Line 76: “With the recently development of OCTA”

Recent development

4. Line 78: “we can easily to get insight”

Please remove “to”

Methods:

1. Line 105: how did the patient present to your clinic? Acute vs chronic PACG? Did you see changes in retinal vessel density (peripapillary vessel densityand macular vessel density) in patient with acute angle closure.

2. How many image did you take for each patient at every visit, how many operators? Did you check for repeatability and reproducibility for image quality control?

Result:

1. Line 156: you can mention the mean of age for men and women, and remove “we could see that there was no statistically significant difference between normal and glaucoma groups for age and gender”

2. Line 161: Table 2 : can you please align all numbers in table 2? For example, each column should start at a certain point and all numbers in this column should start at same point and use separate cells. Here is the journal guidelines for table: https://journals.plos.org/plosone/s/tables .

3. Line 184: “The data suggested correspondence” I believe you mean correlation.

4. Line 194: Table 3 : same for table 2.

Reviewer #2: Is the manuscript presented in an intelligible fashion and written in standard English?

Line 47: I would remove "As" and being the sentence with "In addition"

Line 52: I would replace "while" with "and"

Line 54: I would replace "suggested" with "postulate" or "theorize"

Line 54: I would replace "might" with "may"

Line 55: "facilitated" should be changed to "facilitate"

Line 63: Change "were damaged" to "suffer from"

Line 63: Change "As we all known" to "It is well documented that"

Line 64: Change "monitor" to "monitoring"

Line 65: I would change "visual" to "vision"

Line 67: Change "all known" to "know"

Line 68: Change "is" to "occurs"

Line 68: Remove "some customs"

Line 69: Change "standard" to "standards"

Line 70: End the Glaucoma with a period and not a comma

Line 71: Change "had been" to "is"

Line 73: Change "Reducing" to "reduction"

Line 75: Remove "in the disease"

Line 77: Remove "various of" and change sentence to "in various retinal regions"

Line 78: Change "to get" to "obtain"

Line 79: remove "so"

Line 83: change "also had" to "has"

Line 84: Change "and correlates" to "correlating"

Line 84: Replace "But these researches" to "These studies however"

Line 85 and 86: I would replace "Because of the anatomy various, the number" to "Due to anatomical variations, the prevalence"

Line 86: Change "are much more" to "is greater"

Line 88: I would add the word "and" in between "PACG" and "to evaluate"

Line 92: I would change "study at department" to "study at the department"

Line 112: I would change "reliably" to "reliable"

Line 123: I would add the word "was" before "based" and replace "decide" with "determined"

Line 124: I would replace "severe" with "severity"

Line 138: I would replace "to analyze" with "for analysis"

Line 156: I would replace "we could see" with "demonstrates"

Line 157: I would change "Compared to normal group" to "Compared to the normal group"

Line 184: I would add "was appreciated" after (figure 1c)

Line 188: I would replace "was" with "had"

Line 191: I would replace "agree" with "in agreement"

Line 194: Replace "showed" with "highlighted"

Line 196: Replace "were strongly positively correlated" "showed a strong positive correlation"

Line 202: Replace "area of macular" with "areas of the macula"

The sentence benign at line 230 to line 234 is not very clear and I would advise restricting/rewording

Please reword the sentence starting at Line 242

Additional comments:

1) Could you explain why 20 eyes rather than 22 eyes were used for the 11 normal subjects and why 198 eyes rather than 210 eyes were sued for the 105 PACG patients?

6. PLOS authors have the option to publish the peer review history of their article (what does this mean?). If published, this will include your full peer review and any attached files.

Reviewer #1: No

Reviewer #2: Yes: Mike Zein, MD

---

## [Author Response · Author response to Decision Letter 0]

24 Mar 2020

Dear editor and reviewers:

 It is with excitement that I resubmit to you a revised version of manuscript “Comparisons of retinal vessel density and glaucomatous parameters in optical coherence tomography angiography” for the "PLOS ONE ". Thank you for your guidance in revising my manuscript and giving me the opportunity to resubmit this manuscript. We are sending the revised manuscript according to the comments of the reviewers. Revised portion are rectified in red in “Revised Manuscript with Track Changes”. We have responded specifically to each suggestion below, beginning with your own. To make the changes easier to identify where necessary, I have numbered them.

Journal Requirements:

Response:

As suggested by editor, I have rectified my manuscript to meet “PLOS ONE’s style requirements in revised manuscript. 

2. We noticed you have some minor occurrence of overlapping text with the following previous publication(s), which needs to be addressed.

 Response:

 As suggested by editor, I have rewritten some parts of my manuscript and cited all sources might be duplicated text in my revised manuscript.

3. We note that you have indicated that data from this study are available upon request. PLOS only allows data to be available upon request if there are legal or ethical restrictions on sharing data publicly. 

Response:

I’m pleasure to share my data referred to this manuscript, I have uploaded the data as supporting information files to replicate my study findings. 

Please remove any funding-related text from the manuscript and let us know how you would like to update your Funding Statement. 

Response:

As suggested by editor, I have removed any funding-related text from the manuscript and updated my Finding statement.

5. PLOS requires an ORCID iD for the corresponding author in Editorial Manager on papers submitted after December 6th, 2016. Please ensure that you have an ORCID iD and that it is validated in Editorial Manager. 

Response:

As suggested by editor, I have updated my ORCID ID information.

Review Comments to the Author

Reviewer #1: 

1. Abstract:

Purpose:

1). Line 29: what do you mean by administrative abilities?

Response:

Following the suggestion of the reviewer, I have replaced “administrative” with “monitoring”( Line 29,Page 2)

Results:

2)Line 45-46: “Reduced peripapillary and macular vessel density in glaucoma was detected and a statistically significant correlation with glaucoma stages (P<0.01).” correction: were

Response:

Following the suggestion of the reviewer, I have corrected the mistake. (Line 46, page 2) .

Conclusion:

3). Line 51-55: need English grammar correction, I will reword the conclusion

Response”

 Following the suggestion of the reviewer, I have rewritten the conclusion. (Line 56-62, Page 3)

2. Introduction:

1). Line 63: “As all we known” you mean as all we know?

Response:

Following the suggestion of the reviewer, I have corrected the mistake. (Line 70-71, Page 3)

2). Line 67: as all we know

Response:

Following the suggestion of reviewer, I have corrected the grammar mistake. (Line 76, Page 3)

3). Line 76: “With the recently development of OCTA” Recent development

Response:

Following the suggestion of reviewer, I have corrected the grammar mistake.(Line 86, Page 4)

4). Line 78: “we can easily to get insight” Please remove “to”

 Response:

Following the suggestion of reviewer, I have corrected the grammar mistake.(Line 89, Page 4)

3. Methods: 

 1) Line 105: how did the patient present to your clinic? Acute vs chronic PACG? Did you see changes in retinal vessel density (peripapillary vessel densityand macular vessel density) in patient with acute angle closure.

Response:

Most patients complained of blurred vision, but when we found increased intraocular pressure and changes in optic disc, we will conduct further examination for these suspected glaucoma patients. In this manuscript, all cases were chronic PACG. Acute angle-closure glaucoma is usually accompanied by corneal edema, which makes it difficult to obtain clear OCTA images. Therefore, we did not observe the changes of OCTA in patients with acute angle-closure.

 2). How many images did you take for each patient at every visit, how many operators? Did you check for repeatability and reproducibility for image quality control?

 Response:

 We have two operators, one is responsible for OCTA scanning, the other one is responsible for VF examination. In order to get best quality images, we usually take 3 OCTA images for macular and 3 for optic disc for each patient at every visit. I am responsible for checking the repeatability and reproducibility for all images. The best one was selected for statistical analysis.

4. Result:

. 1) Line 156: you can mention the mean of age for men and women, and remove “we could see that there was no statistically significant difference between normal and glaucoma groups for age and gender”

Response:

Following the suggestion of the reviewer, I have made some changes in my revised manuscript.(Line 169, Page 6)

2) Line 161: Table 2 : can you please align all numbers in table 2? For example, each column should start at a certain point and all numbers in this column should start at same point and use separate cells. Here is the journal guidelines for table: https://journals.plos.org/plosone/s/tables .

 Response:

Following the suggestion of the reviewer, I have aligned all numbers in table 2. 

3). Line 184: “The data suggested correspondence” I believe you mean correlation.

Response:

Following the suggestion of the reviewer, I have made some modifications to this sentence.(Line 199, Page 7)

4). Line 194: Table 3 : same for table 2.

Response:

Following the suggestion of the reviewer, I have made some modifications to table 3.

Reviewer #2: 

1) Line 47: I would remove "As" and being the sentence with "In addition"

Response:

Following the suggestion of reviewer, I have removed “As” and being the sentence with “In addition”.(Line 47, Page 2)

2) Line 52: I would replace "while" with "and"

Response:

Following the suggestion of reviewer, I have replaced “while” with “and”.(Line 56, Page 3)

3) Line 54: I would replace "suggested" with "postulate" or "theorize"

Response:

Following the suggestion of reviewer, I have replaced “suggested ” with “theorize ”.( Line 61, Page 3)

4) Line 54: I would replace "might" with "may"

Response:

Following the suggestion of reviewer, I have replaced “might ” with “may ”.(Line 61, Page 3)

5) Line 55: "facilitated" should be changed to "facilitate"

Response:

Following the suggestion of reviewer, I have changed “facilitated” to “facilitate”.

6) Line 63: Change "were damaged" to "suffer from"

Response: 

Following the suggestion of reviewer, I have changed “were damaged” to “suffer from”.(Line 70, Page 3)

7) Line 63: Change "As we all known" to "It is well documented that" 

Response:

Following the suggestion of reviewer, I have changed “As we all known” to “It is well documented that”. (Line 70-71, Page 3)

8) Line 64: Change "monitor" to "monitoring"

Response:

Following the suggestion of reviewer, I have changed “monitor” to “monitoring”.(Line 71, Page 3)

9 ) Line 65: I would change "visual" to "vision"

Response:

Following the suggestion of reviewer, I have changed “visual” to “vision”.(Line 72, Page 3)

10) Line 67: Change "all known" to "know"

Response:

Following the suggestion of reviewer, I have changed “all known” to “know”. (Line 76, Page 3)

11) Line 68: Change "is" to "occurs"

Response:

Following the suggestion of reviewer, I have changed “is” to “occurs”. (Line 77, Page 3)

12) Line 68: Remove "some customs"

Response:

Following the suggestion of reviewer, I have removed “some customs”.(Line 77, Page 3)

13) Line 69: Change "standard" to "standards"

Response:

Following the suggestion of reviewer, I have changed “standard” to “standards”.(Line 78, Page 3)

14) Line 70: End the Glaucoma with a period and not a comma

Response:

Following the suggestion of reviewer, I have changed comma to period. (Line 79, Page 3)

15) Line 71: Change "had been" to "is"

Response:

Following the suggestion of reviewer, I have changed “had been” to “is”.(Line 80, Page 3)

16) Line 73: Change "Reducing" to "reduction"

Response:

Following the suggestion of reviewer, I have changed “Reducing” to “reduction”(Line 82, Page 3)

17) Line 75: Remove "in the disease"

Response:

Following the suggestion of reviewer, I have removed “in the disease”.(Line 84-85, Page 3)

18) Line 77: Remove "various of" and change sentence to "in various retinal regions"

Response:

Following the suggestion of reviewer, I have removed “various of “and changed sentence to “in various retinal regions”. (Line 87, Page 4)

19) Line 78: Change "to get" to "obtain"

Response:

Following the suggestion of reviewer, I have changed “to get” to “obtain”. (Line 89, Page 4)

20) Line 79: remove "so"

Response:

Following the suggestion of reviewer, I have removed “so”.(Line 90, Page 4)

21) Line 83: change "also had" to "has".

Response:

Following the suggestion of reviewer, I have changed “also had” to “has”. (Line 93, Page 4)

22) Line 84: Change "and correlates" to "correlating”.

Response:

Following the suggestion of reviewer, I have changed “and correlates” to “correlating”.(Line 94, Page 4)

23) Line 84: Replace “But these researches” to “These studies however ".

 Response:

Following the suggestion of reviewer, I have replaced “But these researches” to “These studies however ".(Line 95 ,Page 4)

24) Line 85 and 86: I would replace "Because of the anatomy various, the number" to "Due to anatomical variations, the prevalence"

Response:

Following the suggestion of reviewer, I have replaced Because of the anatomy various, the number" to "Due to anatomical variations, the prevalence".(Line 97, Page 4)

25) Line 86: Change "are much more" to "is greater"

Response:

Following the suggestion of reviewer, I have changed "are much more" to "is greater" (Line 98, Page 4)

26) Line 88: I would add the word "and" in between "PACG" and "to evaluate".

Response:

Following the suggestion of reviewer, I have added the word "and" in between "PACG" and "to evaluate". (Line 99, Page 4)

27) Line 92: I would change "study at department" to "study at the department"

Response:

Following the suggestion of reviewer, I have changed "study at department" to "study at the department". (Line 103, Page 4)

28) Line 112: I would change "reliably" to "reliable"

Response:

Following the suggestion of reviewer, I have changed "reliably" to "reliable".(Line 123, Page 5)

29) Line 123: I would add the word "was" before "based" and replace "decide" with "determined". 

Response:

Following the suggestion of reviewer, I have added the word "was" before "based" and replaced "decide" with "determined". (Line 135-136, Page 5)

30) Line 124: I would replace "severe" with "severity".

Response:

Following the suggestion of reviewer, I have replaced "severe" with "severity". (Line 136, Page 5)

31) Line 138: I would replace "to analyze" with "for analysis"

Response:

Following the suggestion of reviewer, I have replaced "to analyze" with "for analysis". (Line 151, Page 6)

32) Line 156: I would replace "we could see" with "demonstrates".

Response:

Following the suggestion of reviewer 1#, I have made some changed to this paragraph. (Line 169-171, Page 6)

33) Line 157: I would change "Compared to normal group" to "Compared to the normal group".

Response:

Following the suggestion of reviewer, I have changed "Compared to normal group" to "Compared to the normal group". (Line 170, Page 6)

34) Line 184: I would add "was appreciated" after (figure 1c)

Response:

Following the suggestion of reviewer, I have added "was appreciated" after (figure 1c) ( Line 197, Page 6)

35) Line 188: I would replace "was" with "had".

Response:

Following the suggestion of reviewer, I have replaced "was" with "had". (Line 203, Page 6)

36) Line 191: I would replace "agree" with "in agreement".

Response:

Following the suggestion of reviewer, I have replaced "agree" with "in agreement". (Line 206, Page 7)

37) Line 194: Replace "showed" with "highlighted".

Response:

Following the suggestion of reviewer, I have replaced "showed" with "highlighted". (Line 209, Page 8)

38) Line 196: Replace "were strongly positively correlated" "showed a strong positive correlation"

Response:

Following the suggestion of reviewer, I have replaced "were strongly positively correlated" with "showed a strong positive correlation". (Line 211-212, Page 8)

39) Line 202: Replace "area of macular" with "areas of the macula".

Response:

Following the suggestion of reviewer, I have replaced "area of macular" with "areas of the macula". (Line 218, Page 8)

40) The sentence benign at line 230 to line 234 is not very clear and I would advise restricting/rewording.

Response:

Following the suggestion of reviewer, I have reworded the sentence.(Line 250-253, Page 9)

41) Please reword the sentence starting at Line 242

Response:

Following the suggestion of reviewer, I have reworded the sentence.(Line 262, Page 9)

Additional comments:

Could you explain why 20 eyes rather than 22 eyes were used for the 11 normal subjects and why 198 eyes rather than 210 eyes were sued for the 105 PACG patients?

Response:

Our exclusion criteria were presence of any retinal or macular pathology, any media opacities, and any systemic or neurological conditions that could induce unreliable VF testing or poor quality OCTA images. So, some of our subjects had only one eye that met our inclusion criteria for statistical analysis.

Thanks for the help of reviewers and editor, I hope that the revised manuscript is now suitable for publication. Certainly, we are pleased to get more advice from you.

Best wish.

 Li Zhen

---

## [Decision Letter · Decision Letter 1]

18 May 2020

PONE-D-20-04584R1

Comparisons of retinal vessel density and glaucomatous parameters in optical coherence tomography angiography

PLOS ONE

Dear Dr. Li,

Thank you for submitting your manuscript to PLOS ONE. After careful consideration, we feel that it has merit but does not fully meet PLOS ONE’s publication criteria as it currently stands. Therefore, we invite you to submit a revised version of the manuscript that addresses the points raised during the review process.

A learned reviewer has suggested a few minor revision that can be incorporated during this manuscript revision. 

We would appreciate receiving your revised manuscript by Jul 02 2020 11:59PM. To enhance the reproducibility of your results, we recommend that if applicable you deposit your laboratory protocols in protocols.io, where a protocol can be assigned its own identifier (DOI) such that it can be cited independently in the future. For instructions see: http://journals.plos.org/plosone/s/submission-guidelines#loc-laboratory-protocols

We look forward to receiving your revised manuscript.

Kind regards,

Sanjoy Bhattacharya

Academic Editor

PLOS ONE

Reviewers' comments:

Reviewer's Responses to Questions

**Comments to the Author**

1. If the authors have adequately addressed your comments raised in a previous round of review and you feel that this manuscript is now acceptable for publication, you may indicate that here to bypass the “Comments to the Author” section, enter your conflict of interest statement in the “Confidential to Editor” section, and submit your "Accept" recommendation.

Reviewer #2: All comments have been addressed

Reviewer #3: All comments have been addressed

2. Is the manuscript technically sound, and do the data support the conclusions?

Reviewer #2: Yes

Reviewer #3: Yes

3. Has the statistical analysis been performed appropriately and rigorously? 

Reviewer #2: Yes

Reviewer #3: Yes

4. Have the authors made all data underlying the findings in their manuscript fully available?

Reviewer #2: Yes

Reviewer #3: Yes

5. Is the manuscript presented in an intelligible fashion and written in standard English?

Reviewer #2: Yes

Reviewer #3: No

6. Review Comments to the Author

Reviewer #2: (No Response)

Reviewer #3: The content of this manuscript is now sound and fine, though not particularly novel findings. There are a lot of formatting issues. For example, there are non-English characters present in the figures. And also, other font is used among Times New Roman in the main text, which looks messy. Systematic language and formatting editing are needed before being published. Lastly, age typically only needs one digit after the decimal point and correlation only needs two digits after the decimal.

7. PLOS authors have the option to publish the peer review history of their article (what does this mean?). If published, this will include your full peer review and any attached files.

Reviewer #2: Yes: Mike Zein, MD

Reviewer #3: No

---

## [Author Response · Author response to Decision Letter 1]

21 May 2020

Dear editor and reviewers:

 It is with excitement that I resubmit to you a revised version of manuscript “Comparisons of retinal vessel density and glaucomatous parameters in optical coherence tomography angiography” for the "PLOS ONE ". Thank you for your guidance in revising my manuscript and giving me the opportunity to resubmit this manuscript. We are sending the revised manuscript according to the comments of the reviewers. Revised portion are rectified in red in “Revised Manuscript with Track Changes”. As suggested by Reviewer #3, I have made some changes to my figures and tables (Figure 1c, Figure 2b and Table 3), ages have been changed to remain one digit after the decimal point. After I referred the expression of correlation in many references, I found that Pearson’s correlation coefficients were usually kept three digits after the decimal, so, I did not change anymore. In order to looks more clarity, I did some format changes in Table 3.

Thanks for the help of reviewers and editor, I hope that the revised manuscript is now suitable for publication. Certainly, we are pleased to get more advice from you.

Best wish.

 Li Zhen

---

## [Decision Letter · Decision Letter 2]

3 Jun 2020

Comparisons of retinal vessel density and glaucomatous parameters in optical coherence tomography angiography

PONE-D-20-04584R2

Dear Dr. Li,

We’re pleased to inform you that your manuscript has been judged scientifically suitable for publication and will be formally accepted for publication once it meets all outstanding technical requirements.

Kind regards,

Sanjoy Bhattacharya

Academic Editor

PLOS ONE

Additional Editor Comments (optional):

Reviewers' comments:

Reviewer's Responses to Questions

**Comments to the Author**

1. If the authors have adequately addressed your comments raised in a previous round of review and you feel that this manuscript is now acceptable for publication, you may indicate that here to bypass the “Comments to the Author” section, enter your conflict of interest statement in the “Confidential to Editor” section, and submit your "Accept" recommendation.

Reviewer #2: All comments have been addressed

Reviewer #3: All comments have been addressed

2. Is the manuscript technically sound, and do the data support the conclusions?

Reviewer #2: Yes

Reviewer #3: Yes

3. Has the statistical analysis been performed appropriately and rigorously? 

Reviewer #2: Yes

Reviewer #3: Yes

4. Have the authors made all data underlying the findings in their manuscript fully available?

Reviewer #2: Yes

Reviewer #3: Yes

5. Is the manuscript presented in an intelligible fashion and written in standard English?

Reviewer #2: Yes

Reviewer #3: Yes

6. Review Comments to the Author

Reviewer #2: (No Response)

Reviewer #3: My concerns are addressed. Note that, before the emergence of OCTA, there are some research which have investigated if and how glaucoma is related to major retinal vessel locations. See the following and please cite these works in the Introduction:

Baniasadi, N., Wang, M., Wang, H., Mahd, M. and Elze, T., 2017. Associations between optic nerve head–related anatomical parameters and refractive error over the full range of glaucoma severity. Translational vision science & technology, 6(4), pp.9-9.

Radcliffe, N.M., Smith, S.D., Syed, Z.A., Park, S.C., Ehrlich, J.R., De Moraes, C.G., Liebmann, J.M. and Ritch, R., 2014. Retinal blood vessel positional shifts and glaucoma progression. Ophthalmology, 121(4), pp.842-848.

7. PLOS authors have the option to publish the peer review history of their article (what does this mean?). If published, this will include your full peer review and any attached files.

Reviewer #2: Yes: Mike Zein, MD

Reviewer #3: No

---

## [Editor Report · Acceptance letter]

5 Jun 2020

PONE-D-20-04584R2 

Comparisons of retinal vessel density and glaucomatous parameters in optical coherence tomography angiography 

Dear Dr. Li:

I'm pleased to inform you that your manuscript has been deemed suitable for publication in PLOS ONE. Congratulations! Your manuscript is now with our production department. 

Kind regards, 

on behalf of

Dr. Sanjoy Bhattacharya 

Academic Editor

PLOS ONE